# From Gut to Brain: Uncovering Potential Serum Biomarkers Connecting Inflammatory Bowel Diseases to Neurodegenerative Diseases

**DOI:** 10.3390/ijms25115676

**Published:** 2024-05-23

**Authors:** Oliviu-Florentiu Sarb, Adriana-Daniela Sarb, Maria Iacobescu, Irina-Maria Vlad, Mircea-Vasile Milaciu, Lorena Ciurmarnean, Vitalie Vacaras, Alina-Ioana Tantau

**Affiliations:** 1Department of Neuroscience, Iuliu Hatieganu University of Medicine and Pharmacy, 400012 Cluj-Napoca, Romania; sarboliviu@yahoo.com (O.-F.S.); irina.vlad0001@gmail.com (I.-M.V.); 2Department of Internal Medicine, 4th Medical Clinic, Iuliu Hatieganu University of Medicine and Pharmacy, 400012 Cluj-Napoca, Romania; mircea_milaciu@yahoo.com (M.-V.M.); lorena_ciumarnean@yahoo.com (L.C.); alitantau@gmail.com (A.-I.T.); 3Department of Internal Medicine, Heart Institute, Iuliu Hatieganu University of Medicine and Pharmacy, 400012 Cluj-Napoca, Romania; adriana_porca@yahoo.com; 4Department of Proteomics and Metabolomics, MEDFUTURE Research Center for Advanced Medicine, Iuliu Hatieganu University of Medicine and Pharmacy, 400012 Cluj-Napoca, Romania; ilies.maria@umfcluj.ro

**Keywords:** inflammatory bowel diseases, Crohn’s Disease, ulcerative collitis, neurodegeneration, biomarkers, gut-brain axis, gut microbiome, Alzheimer’s disease, Parkinson’s disease, mild cognitive impairment

## Abstract

Inflammatory bowel diseases (IBDs) are characterized by chronic gastrointestinal inflammation due to abnormal immune responses to gut microflora. The gut–brain axis is disrupted in IBDs, leading to neurobiological imbalances and affective symptoms. Systemic inflammation in IBDs affects the brain’s inflammatory response system, hormonal axis, and blood–brain barrier integrity, influencing the gut microbiota. This review aims to explore the association between dysregulations in the gut–brain axis, serum biomarkers, and the development of cognitive disorders. Studies suggest a potential association between IBDs and the development of neurodegeneration. The mechanisms include systemic inflammation, nutritional deficiency, GBA dysfunction, and the effect of genetics and comorbidities. The objective is to identify potential correlations and propose future research directions to understand the impact of altered microbiomes and intestinal barrier functions on neurodegeneration. Serum levels of vitamins, inflammatory and neuronal damage biomarkers, and neuronal growth factors have been investigated for their potential to predict the development of neurodegenerative diseases, but current results are inconclusive and require more studies.

## 1. Introduction

### 1.1. Inflammatory Bowel Diseases

Inflammatory bowel diseases (IBDs) are chronic conditions characterized by repetitive episodes of inflammation in the gastrointestinal tract due to an abnormal immune response to gut microflora. The two main types of IBDs are Crohn’s disease (CD) and ulcerative colitis (UC). UC involves diffuse inflammation of the colonic mucosa, while CD results in transmural ulceration of any portion of the gastrointestinal tract, most often affecting the terminal ileum and colon [1]. IBDs occur at any age, with peaks at 15–35 years and after 60 years. It equally affects males and females. The highest prevalence rates are reported in Europe and North America, with values exceeding 0.3% in these regions. The incidence and prevalence of IBDs are increasing in newly industrialized countries in Africa, Asia, and South America, indicating the emergence of IBDs as a global disease. The incidence of IBDs is stabilizing in Western countries, while it continues to rise in developing nations. Environmental factors, such as urbanization and westernized lifestyles, are believed to contribute to the increasing incidence of IBDs [2].

The main pathogenic changes in IBDs involve a combination of genetic, environmental, and immunological factors [3,4,5].
There is an overactive immune response in the gut, leading to inflammation and damage to the intestinal tissue. This immune response involves various immune cells, including T cells, B cells, and macrophages.The imbalance of gut bacteria is present in IBDs, leading to inflammation and immune activation.Genetic factors have been associated with an increased risk of developing IBDs. These genetic factors can affect the immune system, barrier functions of the gut, and the body’s response to inflammation.Environmental factors, such as diet, smoking, and infections, can play a role in triggering or exacerbating IBDs. These factors can interact with the gut microbiota and the immune system, contributing to inflammation.

### 1.2. Gut–Brain Axis

The gut–brain axis (GBA) refers to the bidirectional communication between the gut and the brain, which involves neural, hormonal, and immune pathways. This communication allows the brain to influence intestinal functions, including the gut microbiota, and vice versa [6]. The gut–brain–microbiome axis (GBMA) is another term that also involves gut bacteria in this definition.

IBDs are associated with systemic inflammation and alterations of the GBA. In IBDs, the GBA is disrupted, leading to neurobiological imbalances and clinical affective and/or behavioral symptoms [7]. Systemic inflammation can also activate the inflammatory response system in the brain, the hypothalamic–pituitary–adrenal axis (HPAA), and brain areas implicated in altered behaviors. This can lead to changes in blood–brain barrier (BBB) integrity, and it might explain the emerging role of gut microbiota, and the clinical response to probiotics in IBDs [8]. Inflammation can also disrupt the balance of the gut microbiota, a condition known as dysbiosis. Dysbiosis affects the GBA and might lead to changes in brain function, resulting in an altered HPAA [6,9]. This gives rise to an overall pro-inflammatory phenotype and dysregulated HPAA and serotonergic functioning [10]. Additionally, the gut microbiota produces various metabolites, such as short-chain fatty acids (SCFAs), which can influence brain function. Changes in the production of these metabolites due to dysbiosis could therefore impact the brain [6].

In summary, systemic inflammation and alterations in the GBA play a significant role in the pathophysiology of IBDs, affecting not only the gastrointestinal tract but also the brain and behavior [6,7,8,9,10].

### 1.3. Neurodegenerative Diseases

Neurodegenerative diseases are a group of chronic and progressive disorders characterized by the gradual and progressive loss of neurons in the central nervous system (CNS) or peripheral nervous system (PNS). These diseases affect specific regions of the brain or spinal cord, leading to a decline in cognitive function, movement, and other neurological functions. Examples of neurodegenerative diseases include Alzheimer’s disease (AD), Parkinson’s disease (PD), Huntington’s disease (HD), amyotrophic lateral sclerosis (ALS), frontotemporal lobe degeneration (FTD), and multisystem atrophy (MSA) [11]. There are some authors that claim that multiple sclerosis (MS) is a neurodegenerative disease, rather than an autoimmune disease [12]. These diseases are often associated with the accumulation of abnormal proteins in the brain, leading to the degeneration and death of neurons. Currently, there is no cure for neurodegenerative diseases, and treatments focus on managing symptoms and slowing disease progression [11,13,14,15].

Neurodegenerative diseases that primarily affect cognitive function, such as AD, are characterized by specific pathological and histological changes in the brain. These include the accumulation of abnormal proteins, such as amyloid-beta plaques and tau tangles, which disrupt neuronal function and lead to neuronal loss. Also, there might be impaired protein clearance. In addition, there may be inflammation, gliosis, mitochondrial dysfunction, and synaptic dysfunction. These changes are often accompanied by atrophy and shrinkage of specific brain regions, particularly those involved in memory and cognition. The precise pattern and distribution of these changes vary depending on the specific neurodegenerative disease [16,17,18]. The prevalence rate in the Eastern Mediterranean Region of AD and other dementias was estimated at 759.8 per 100,000, and for PD, it was estimated at 87.1 per 100,000 [19]. In other studies, the prevalence rate was 1.17% for the 65–69 age group, while a rate of 54.83% was observed for those aged 95 years or more [20].

Mild cognitive impairment (MCI) is a syndrome characterized by cognitive decline that is greater than expected for an individual’s age and education level but does not significantly interfere with daily activities. It is considered a transitional stage between normal aging and dementia [21]. The prevalence of MCI varies depending on the population studied and the diagnostic criteria used. Prevalence estimates range from 3% to 29.9% in adults older than 60 years. The incidence rates of MCI range from 9.9 to 76.8 per 1000 person-years [22,23,24]. MCI is more prevalent in older individuals and those with lower education levels. It is associated with an increased risk of developing dementia, particularly AD [24]. The risk factors for MCI are increasing age, comorbidities (diabetes, high blood pressure, obesity, obstructive sleep apnea, and hypercholesterolemia), lifestyle risk factors (smoking, alcohol consumption, depression, and lack of social relationships), vascular risk factors (hyperlipidemia, hypercholesterolemia, and hyperhomocysteinemia), genetic factors, and chronic diseases associated with systemic inflammation [21,25,26,27,28,29].

### 1.4. Gut–Brain Axis, Intestinal Microbiome, and Neurodegenerative Diseases

There is growing evidence that the gut microbiota is closely related to the development and progression of neurodegenerative diseases. Multiple studies have shown that alterations in the gut microbiota, known as gut dysbiosis, may contribute to the pathogenesis of these diseases. The gut microbiota can influence the CNS through various mechanisms. It can modulate the immune system, produce molecules and metabolites that affect the brain, and regulate the integrity of the intestinal and blood–brain barriers. Dysbiosis of the gut microbiota may disrupt these processes and contribute to neuroinflammation, protein misfolding, and other pathological features observed in neurodegenerative diseases [30,31,32].

Although there is still much to learn about the precise mechanisms underlying the gut–brain axis in neurodegenerative diseases, studies have shown promising outcomes in manipulating the gut microbiota for therapeutic purposes. Clinical and preclinical trials have explored interventions such as dietary changes, probiotic and prebiotic supplementation, and fecal microbiota transplantation. These interventions aim to restore a healthy gut microbiota and potentially mitigate cognitive decline in neurodegenerative diseases. However, further research is needed to fully understand the therapeutic potential and optimize these interventions [33,34,35].

## 2. Aim

Considering that nearly 2500 years ago, Hippocrates had a dazzling statement declaring that ‘all disease begins in the gut’ without having any of our modern clinical and laboratory exploration techniques, we strongly believe that it is time that we find a clear theory and mechanism of how the altered microbiome and intestinal barrier dysfunction might lead to the development of neurodegenerative diseases [10,31,33,36].

The purpose of this review is to identify some of the possible serum biomarkers that could be useful clinical tools to predict the development of neurodegenerative diseases that primarily affect cognition among IBD patients and determine the perspectives we could focus on in the near future to test those hypotheses. Knowing this, further research could focus more on the intestinal microbiome and other mechanisms and determine how we could influence those presumed pathological pathways to reduce the global burden of dementia. Also, despite all the research carried out around the globe, the future perspectives of a patient diagnosed with dementia are currently discouraging. Identifying a patient with MCI could prove useful in preventing further development of dementia. It is worth mentioning that due to an increase in life expectancy during the last 100 years, and given that aging is known to be a risk factor for neurodegeneration and dementia, our common goal should be to work towards the prevention of these diseases to decrease the global caregiving costs for patients with cognitive disabilities [16,37,38,39].

Serum biomarkers could prove useful because of their relatively low cost and because they do not require any other invasive methods such as lumbar puncture or a more expensive investigation such as a functional brain MRI. Their dual role found in both IBDs and neurodegenerative diseases could also be useful for risk stratification and choosing the right medication for each patient. 

## 3. Is Neurodegeneration an Extraintestinal Manifestation of IBDs?

Several studies have investigated the association between IBDs and neurodegenerative diseases. A study conducted in Korea found that IBD patients had a higher risk of developing PD and AD compared to non-IBD controls [40]. Another study using Mendelian randomization analysis did not find clear evidence for a causal association between IBDs and PD or other neurodegenerative disorders [41]. However, a meta-analysis of cohort studies revealed an increased risk of stroke, all-cause dementia, PD, and MS in IBD patients [42]. The GBA and gut microbiota dysbiosis are thought to play a role in the association between IBDs and neurodegenerative diseases [9,43,44,45,46].

The pathogenic mechanisms linking IBDs to neurodegenerative diseases are not fully understood. Several potential mechanisms have been proposed based on available evidence. Several molecular and cellular pathological changes at the CNS level were also speculated. In summary, potential mechanisms and changes at the CNS level are as follows:Shared genetic loci were identified between IBDs and neurodegenerative diseases. For example, the LRRK2 gene has been implicated in both PD and CD [47].Chronic inflammation is a hallmark of both IBDs and neurodegenerative diseases. Inflammatory mediators and immune cells can cross the BBB and contribute to neuroinflammation, neuronal damage, and neurodegeneration [43,47,48,49]. Increased oxidative stress, resulting from an imbalance between reactive oxygen species (ROS) production and antioxidant defense mechanisms, is implicated in the pathogenesis of both IBDs and neurodegenerative diseases [43]. The integrity of the BBB might be compromised in IBDs, allowing the entry of inflammatory mediators and immune cells into the CNS [43]. Some of the inflammatory blood biomarkers associated with neurodegeneration include cytokines (such as IL-6, IL-8, and TNF-α), chemokines (such as MCP-1), growth factors (such as VEGF), and immune-related proteins (such as ICAM-1 and VCAM-1). These biomarkers have been found to be altered in the serum of patients with AD and MCI [50,51,52]. Elevated levels of C-reactive protein (CRP) have been associated with an increased risk of AD and PD [53,54]. Increased levels of IL-6 have been observed in AD, PD, and other neurodegenerative diseases [54]. TNF-alpha has been implicated in the pathogenesis of AD, PD, and other neurodegenerative diseases [54]. Markers of microglial activation, such as CD68 and HLA-DR, have been associated with neurodegenerative diseases [55]. Elevated levels of IL-1β have been found in the brains of individuals with AD and PD [54].Abnormal protein aggregation is a characteristic feature of many neurodegenerative diseases, such as the accumulation of alpha-synuclein in PD, tau and amyloid-beta in AD, or TDP-43 in frontotemporal lobe degeneration [43,56,57]. It has been suggested that similar protein aggregation processes may occur in the gut of IBD patients, contributing to neurodegenerative pathology [58]. For example, abnormal aggregation of alpha-synuclein has been detected in the enteric neurons of some patients with UC [59].Environmental factors, including gut microbiota composition, diet, and exposure to toxins, may influence the development and progression of both IBDs and neurodegenerative diseases. These factors can interact with genetic susceptibility to modulate disease risk [47]. Nutritional deficits that have been studied in relation to the development of neurodegenerative diseases include deficiencies of essential nutrients such as long-chain polyunsaturated fatty acids, vitamins (such as vitamin E), and mineral elements [60]. Malnutrition and low body mass index (BMI) have also been associated with a higher risk of dementia and mortality [61]. Additionally, chronic over-nutrition and metabolic disorders, such as diabetes, hypertension, dyslipidemia, and atherosclerosis, have been linked to neurodegenerative diseases [62]. Deficiencies of vitamins such as B1, B12, and vitamin D have also been associated with an increased risk of neurodegenerative diseases [63,64,65].Mitochondrial dysfunction, characterized by impaired energy production and increased production of ROS, has been observed in both IBDs and neurodegenerative diseases. The dysfunction of mitochondria can lead to neuronal cell death and contribute to the progression of neurodegeneration [43].The disruption of the neurotrophic factors, which play an important role in the survival, growth, and maintenance of neurons, might exist in both IBDs and neurodegenerative diseases, leading to neuronal dysfunction and degeneration [43].GBA disruption, especially of the intestinal barrier functions, and immune responses in IBDs can lead to systemic inflammation and neuroinflammation, which are associated with neurodegenerative diseases [47,48]. The disruption of the gut–brain axis can affect various brain structures and functions. The insular cortex is involved in processing and integrating sensory information from the gut, such as pain and visceral sensations, and it might be affected by GBA disruption. The cingulate cortex plays a role in regulating emotions and pain perception. Dysfunction in the GBA can influence the functioning of the cingulate cortex, leading to emotional and pain-related disturbances. The hypothalamus is a brain structure involved in regulating various physiological processes, including appetite, metabolism, stress response, and sleep. The disruption of the GBA can also affect hypothalamic function [66,67]. Dysfunction in the GBA has been associated with cognitive impairments, such as difficulties in memory, attention, and executive function. It might also lead to mood disorders, such as anxiety or depression [68]. An increase in neuroinflammation has also been observed in GBA disruption, which might also lead to neurodegeneration [66]. Due to GBA’s effect on the production and regulation of neurotransmitters, such as serotonin and dopamine, its disruption might imbalance these neurotransmitters [68].Adverse effects of therapies such as high-dose corticosteroids have been associated with cognitive impairment, emotional disturbance, and behavioral changes in children and adolescents [69]. However, the use of anti-TNF-α inhibitors is associated with a lower risk of developing AD and PD in patients with IBDs as they may reduce inflammation and modulate the gut microbiome [42,70].

In conclusion, IBDs are pathologies in which the GBA is disrupted and there is an associated systemic inflammation (Figure 1). GBA disruption is a consequence but also a cause of IBDs. Environmental factors might also lead to GBA disruption. Systemic inflammation, which is a main characteristic of IBDs, might lead to abnormal protein aggregation. Genetic factors lead to abnormal protein aggregation, but they are also linked to the development of IBDs. Malabsorption, a complication that occurs in IBD patients, could also lead to the development of neurodegeneration.

Whether neurodegeneration is an extraintestinal manifestation of IBDs still remains a pertinent research question, and future research will hopefully be able to answer this question. GBA disruption and systemic inflammation have a negative effect on CNS health and function. The exact mechanism must be extensively researched. One of the possibilities of research is inquiring the serum biomarkers modified in both IBD and neurodegenerative diseases. Secondly, an objective is to determine whether they act as an active molecule in the process of neurodegeneration or they might serve as predictive biomarkers, so clinicians are given a warning and they should change the treatment of the affected patients to stop the evolution towards a neurodegenerative disease.

Several biomarkers have been chosen by our research team, and they can be categorized as nutritional deficiency biomarkers, neuronal damage biomarkers, and inflammatory biomarkers.

## 4. Nutritional Deficiency Biomarkers

Nutritional deficiencies are common in patients with IBDs. Studies have shown that malnutrition affects a significant proportion of IBD patients, with a prevalence of 25–69% in affected patients [71,72]. These deficiencies can arise from factors such as inadequate dietary intake, malabsorption, and disease activity [73].

Specifically, IBD patients may be deficient in various nutrients, including vitamins and minerals. Some common deficiencies observed in IBD patients include iron, zinc, magnesium, vitamin B12, vitamin D, and calcium [74,75]. These deficiencies can have important consequences for the patient’s health and may lead to complications such as anemia, impaired bone health, and impaired wound healing [76]. Some of the nutritional deficiencies have been linked to the development of neurodegenerative diseases, while others showed no significant effect [6].

Whether the development of neurodegenerative diseases in IBD patients is a direct consequence of nutritional deficiency or only serves as a co-factor remains to be answered.

### 4.1. Vitamin D3

Vitamin D3, also known as cholecalciferol, is a form of vitamin D that is synthesized in the skin when it is exposed to sunlight or obtained from certain foods and supplements. It plays an important role in maintaining calcium and phosphorus levels in the body, which are essential for bone health.

In the immune system, vitamin D3 has been shown to have immunomodulatory effects [77,78]. For example, it can inhibit the differentiation and activation of certain immune cells, such as T cells and dendritic cells, and promote the production of anti-inflammatory cytokines like IL-10, suggesting that vitamin D3 may have a protective role in autoimmune and inflammatory diseases [79,80,81].

In the CNS, vitamin D3 has been found to be synthesized and activated within the brain. It plays a role in gene regulation, inflammation, and cellular proliferation, differentiation, DNA repair, and apoptosis [82,83,84,85]. It regulates genes involved in calcium homeostasis, oxidative stress, immune response, and neuroprotection. For example, it can modulate the activation of microglia, which are immune cells in the CNS, reducing their production of pro-inflammatory cytokines and increasing the expression of anti-inflammatory cytokines like IL-10, suggesting that vitamin D3 may have neuroprotective effects and help prevent immune-mediated damage in the CNS [80,86]. Also, vitamin D3 regulates various processes in the gut–brain axis, including the integrity of the intestinal barrier, the composition of the gut microbiota, and the production of antimicrobial peptides [77,78,87]. It also modulates immune responses in the gut, reducing inflammation and promoting immune tolerance. Also, vitamin D receptors are found in the brain, meaning it might have a direct role in the CNS [84,88]. These effects of vitamin D3 in the gut can have indirect effects on the brain. For example, by maintaining gut health and reducing inflammation, vitamin D3 may help protect against neurodegenerative diseases and improve cognitive function.

In patients with IBDs, vitamin D deficiency is common. Studies have shown that a significant proportion of CD and UC patients have low serum levels of vitamin D3. For example, one study found that 58.6% of CD patients and 44.6% of UC patients had vitamin D3 levels below 50 nmol/L [89]. Another study reported that 53% of CD patients and 44% of UC patients had vitamin D3 levels below 50 nmol/L [90]. Additionally, vitamin D3 levels were found to be lower during periods of disease activity compared to remission in both CD and UC patients [91].

In patients with neurodegenerative diseases, low serum levels of vitamin D3 have been observed. Studies have shown that low serum levels of the circulating form of vitamin D are associated with an increased risk of AD, PD, HD, and MCI [85]. Additionally, vitamin D deficiency has been linked to an increased risk of incident all-cause dementia and AD [85].

### 4.2. Vitamin B12

Vitamin B12, also known as cobalamin, is a vitamin that is necessary for erythropoiesis, the maintenance of a healthy nervous system, and protein and lipid metabolism. Additionally, vitamin B12 contributes to DNA synthesis and supports brain function [92].

In the immune system, vitamin B12 is involved in the regulation of immune cell production and function, helping to maintain a balanced immune response. For example, vitamin B12 deficiency can lead to alterations in lymphocyte populations and natural killer cell activity, which are involved in cytotoxic response [93].

In the CNS, vitamin B12 is important for cellular function, especially to produce and maintain healthy myelin sheaths, which are essential for proper nerve signal transmission and axonal protection [92,94]. Vitamin B12 is also necessary for the synthesis of S-adenosyl methionine, a methyl donor that has a very important function in methylation reactions in the CNS, maintaining the integrity of the myelin sheath and the methylation of myelin basic proteins and lipids [95]. The deficiency of vitamin B12 can lead to neurological symptoms such as myelopathy, neuropathy, dementia, optic neuropathy, and subacute combined degeneration of the spinal cord [92,96,97].

In patients with IBDs, vitamin B12 levels are affected. Studies have shown that the prevalence of vitamin B12 deficiency is higher in patients with CD compared to those with UC. In CD, the prevalence of vitamin B12 deficiency ranges from 15.6% to 33%, while in UC, it ranges from 2.8% to 16% [98]. Risk factors for vitamin B12 deficiency in Crohn’s disease include ileal resection length, ileal inflammation, and disease activity. Patients with shorter ileal resection lengths and active ileal inflammation are more likely to have vitamin B12 deficiency [99]. The length of the resected ileal segment correlates with the degree of vitamin B12 absorption impairment [100]. Although the mean values for vitamin B12 levels do not differ significantly between patients with CD and the controls, there is an increase in the binding capacity of transcobalamin in CD patients, suggesting altered intracellular vitamin B12 status [101].

In neurodegenerative diseases, vitamin B12 levels can be affected. Studies have shown that low vitamin B12 levels are associated with AD, VD, PD, and MCI [27,65,73,102]. In patients with AD, the median serum vitamin B12 concentration was significantly lower compared to the controls [26,64]. In PD, low B12 levels at the time of diagnosis were associated with a higher risk of developing dementia [102].

The presumed mechanisms linking vitamin B12 deficiency and the development of neurodegeneration involve the role of vitamin B12 in DNA synthesis, myelin maintenance, and methylation reactions. Vitamin B12 deficiency disrupts these methylation reactions, leading to damage to the myelin sheath and subsequent neurodegeneration. Additionally, vitamin B12 deficiency can result in the accumulation of abnormal fatty acids and disruption of normal myelin synthesis [95]. Other mechanisms include the imbalance of tumor necrosis factor-alpha (TNF-α) and epidermal growth factor levels, as well as the accumulation of homocysteine and methyl malonyl-CoA [102]. In patients with PD, vitamin B12 supplementation has been found to reduce oxidative stress and improve motor function [103]. A study found that vitamin B12 supplementation improved frontal function in patients with MCI [104]. Another study showed that vitamin B12 treatment improved frontal lobe and language function in patients with MCI, although it rarely reversed dementia [105]. Also, another study suggested that vitamin B12 deficiency may contribute to progressive brain atrophy in elderly individuals [106].

### 4.3. Vitamin B9

Vitamin B9, also known as folate or folic acid, is a water-soluble vitamin that plays a role in various biological processes. Folate is the naturally occurring form found in foods, while folic acid is the synthetic form used in supplements and fortified foods. Folate plays a role in maintaining DNA methylation, which is important for gene expression and chromatin structure. It is also involved in the synthesis of DNA nucleotides and the regeneration of methionine, a key component in methylation reactions. Folate deficiency can lead to DNA damage and impaired regenerative potential of tissues [107].

In the immune system, vitamin B9 is involved in regulating the differentiation of T helper cells, such as Th9 cells, which help maintain immune homeostasis [108].

In the CNS, vitamin B9 is essential for brain development and function. The deficiency of vitamin B9 during early life can have long-term effects on mental health and behavior in adults [109]. Vitamin B9 is also involved in the synthesis of neurotransmitters, such as serotonin, norepinephrine, and dopamine, which are important for mood regulation [110]. Additionally, vitamin B9 deficiency has been associated with psychiatric symptoms, and supplementation with the active form of vitamin B9, L-methyl folate, has been shown to reduce psychiatric symptoms and improve treatment outcomes [110].

There is limited information regarding folate levels in CD and UC patients, but some studies showed that their serum level is reduced [76,111,112]. Also, regarding neurodegenerative disease, the evidence is limited. For example, there are studies that observed a lower level of folate in ALS patients, as well as AD and PD patients [113,114,115].

### 4.4. Homocysteine

Homocysteine (HCY) is an amino acid that is involved in the methylation and sulphuration pathways and is derived from the metabolism of methionine, an essential amino acid. Elevated levels of HCY, known as hyperhomocysteinemia (HHCY), have been associated with several disorders, including cardiovascular disease (CVD), neurodegenerative diseases, autoimmune disorders, and birth defects [116]. The main risk factors for HHCY are increased age, male gender, family history, vitamin B9 or B12 deficiency, impaired kidney function, and lifestyle factors, including smoking, alcohol drinking, and sedentary behavior [117]. Deficiencies of vitamin B9 and B12 can lead to elevated levels of HCY in the blood. Several studies have shown that supplementation with folic acid and vitamin B12 can effectively lower HCY levels. For example, daily supplementation with 0.5–5 mg of folic acid can reduce HCY concentrations by about 25%, and the addition of vitamin B12 (0.5 mg) can produce an additional 7% reduction. The combination of folic acid and vitamin B12 can reduce HCY levels by about 25–33% [118].

In the immune system, studies have shown that HCY can affect T-lymphocyte function. It has been found to promote T cell activation, differentiation, and activation-induced cell death [119]. These effects on immune function may contribute to age-related immune dysfunction and disease pathology.

In the CNS, HCY has been implicated in the pathogenesis of various neurological diseases, including stroke, AD, PD, epilepsy, MS, and ALS [120]. High levels of HCY have been implicated in the pathogenesis of neurodegenerative diseases through several mechanisms. HHCY can exert direct neurotoxic effects, leading to neuronal dysfunction and death. It can induce excitotoxicity, oxidative stress, calcium accumulation, and apoptosis, all of which contribute to neurodegeneration. Vascular dysfunction is associated with HHCY, including endothelial dysfunction, impaired BBB integrity, and increased risk of cerebrovascular diseases. These vascular changes can contribute to neurodegenerative processes by compromising cerebral blood flow and nutrient supply to the brain. Methylation and epigenetic alterations develop when higher levels of HCY are present in the blood, affecting gene expression and neuronal function. Elevated HCY levels can disrupt methylation reactions, leading to altered gene expression patterns and epigenetic modifications that contribute to neurodegeneration. HCY has been shown to promote the accumulation and aggregation of amyloid-beta and tau proteins. HCY can increase the production of amyloid-beta and enhance its neurotoxicity, as well as facilitate tau hyperphosphorylation and aggregation. Also, HCY can impair glutamate uptake and metabolism, leading to increased extracellular glutamate levels. Excessive glutamate can induce excitotoxicity, causing neuronal damage and death, which are implicated in various neurodegenerative diseases. Also, HHCY can promote inflammation and oxidative stress in the brain, contributing to neurodegeneration. HCY can activate inflammatory pathways, increase the production of reactive oxygen species, and impair antioxidant defenses, leading to neuronal damage [120,121,122,123,124].

Patients with IBDs have significantly higher HCY levels compared to healthy individuals [125,126]. Studies have shown that patients with CD have significantly higher levels of HCY compared to healthy controls. The concentration of serum HCY in CD patients was found to be 13.6 +/− 6.5 µmol/L, which was significantly higher than the levels in the controls (9.6 +/− 3.4 µmol/L) [127]. Another study reported that 52% of CD patients had HHCY [128]. Similarly, patients with UC also exhibited elevated levels of HCY. The concentration of serum HCY in UC patients was found to be 15.9 +/− 10.3 µmol/L, which was significantly higher than the levels in the controls (9.6 +/− 3.4 µmol/L) [127]. Mild HHCY was found in 42% of UC patients [129]. Elevated HCY levels in IBD patients may contribute to the increased thrombotic risk observed in these patients [130]. However, it is important to note that HCY levels can be influenced by vitamin deficiencies, medications, and lifestyle factors [101,131,132].

### 4.5. Vitamin B6

Vitamin B6, also known as pyridoxine, is a water-soluble vitamin that is important for energy metabolism and the synthesis of neurotransmitters, hemoglobin, and DNA. It also plays a role in gene expression and normal brain functioning [133,134].

In the immune system, vitamin B6 plays a role in the production and function of immune cells. It aids in the formation of antibodies and the division of immune cells.

In the CNS, vitamin B6 is involved in the synthesis of neurotransmitters such as dopamine, serotonin, and gamma-aminobutyric acid (GABA), which are essential for proper brain function and mood regulation. It also participates in the metabolism of amino acids, which are the building blocks of proteins, and helps in the production of myelin, the protective covering of nerve fibers. The deficiency of vitamin B6 can lead to neurological symptoms such as neuropathies, hyperirritability, and convulsions [94,131,134].

In IBDs, studies have shown a higher prevalence of vitamin B6 deficiency in patients with CD compared to UC [135]. Vitamin B6 deficiency in CD patients has been associated with small bowel lesions, extraintestinal manifestations, and ileal resection.

In neurodegenerative diseases, vitamin B6 deficiency has been implicated in their pathogenesis, respectively, in PD, AD, MS, and ALS [134,136,137]. Animal studies have shown that vitamin B6 restriction can induce oxidative stress, endoplasmic reticulum stress, and apoptotic cell death in the brain, leading to neurodegeneration [138]. Additionally, vitamin B6 deficiency has been associated with abnormal iron deposition and lipid peroxidation, which are some of the characteristic features of neurodegenerative diseases [139].

### 4.6. Vitamin B1

Vitamin B1, also known as thiamine, is a water-soluble vitamin that functions as a coenzyme in the metabolism of carbohydrates and amino acids, with an important role in energy production and the conversion of glucose to ATP. It also participates in the synthesis of neurotransmitters and plays a significant role in the CNS and immune system [140].

In terms of the immune system, thiamine plays a role in supporting immune function. It helps in the production of antibodies. Thiamine deficiency can impair immune function and make individuals more susceptible to infections [140].

In the CNS, thiamine is essential for the proper functioning of nerve cells. It is involved in the synthesis of neurotransmitters. Thiamine deficiency can lead to neurological symptoms such as confusion, memory problems, and nerve damage [141].

In the context of IBDs, vitamin B1 deficiency may occur due to malnutrition or malabsorption, which can result in anemia. The supplementation of vitamin B12 and folic acid may be necessary, especially in patients with CD or those with inflammation or resection of the terminal ileum. Additionally, patients with high or continuous inflammatory activity and frequent steroid therapy in IBDs have an increased risk of low bone mineral density and vitamin D deficiency [142,143].

In neurodegenerative diseases, serum thiamine levels may be diminished. Thiamine deficiency has been associated with AD, PD, and HD. Studies have shown that thiamine-dependent processes and enzymes are reduced in the brains of patients with these diseases [141,144,145]. Thiamine supplementation has been found to have potential therapeutic benefits in neurodegenerative diseases, improving cognitive outcomes and reducing oxidative stress and neuroinflammation. Optimal thiamine serum levels and doses are yet to be established [146].

## 5. Neuronal Damage Biomarkers

Serum neuronal damage biomarkers are proteins released into the bloodstream because of neuronal injury or neurodegeneration. They can be measured in the blood and used as indicators of neuronal damage in various neurological disorders [147,148]. These biomarkers have clinical utility in several ways. For example, they can aid in the diagnosis of neurological diseases, provide information about disease severity and progression, monitor treatment efficacy, and predict clinical outcomes. Furthermore, serum biomarkers can be used in clinical trials to assess the effects of therapeutic interventions and guide treatment decisions. They provide a minimally invasive and easily accessible method for monitoring neuronal damage and evaluating disease activity [149,150]. Although these biomarkers show promise, further research is needed to fully understand their clinical applications, establish standardized measurement techniques, and determine their optimal use in different neurological disorders [151]. One of the several known neuronal damage biomarkers was also tested in IBD patients. One of the trickiest issues with neuronal damage biomarkers is the lesions that affect the enteric nervous system (ENS).

ENS is a complex network of neurons and glial cells that is located within the walls of the gastrointestinal tract. It is often referred to as the “second brain” because it can function independently of the CNS. The ENS controls various functions of the GI tract, including motility (movement of food), secretion (release of digestive enzymes), and blood flow. It plays a crucial role in regulating gut reflexes and coordinating gut functions. The ENS is involved in gastrointestinal disorders and developmental disabilities [152,153].

In IBDs, the ENS can become altered and dysfunctional, contributing to the pathophysiology and symptoms of the disease. The ENS interacts with the immune system and can modulate aspects of intestinal inflammation through the secretion of neuropeptides. Neuropeptides such as substance P, corticotropin-releasing hormones, neurotensin, vasoactive intestinal peptide, and others are thought to be involved in the pathogenesis of IBDs. Changes in the ENS can affect gut motility, secretion, sensory perception, and immune function, leading to gastrointestinal dysfunction associated with IBDs [154,155,156].

The dysfunction of the ENS may lead to gastrointestinal symptoms commonly seen in neurodegenerative diseases, such as constipation and dysphagia. Additionally, the ENS may be involved in the propagation of abnormal protein accumulation and neuroinflammation through the release of extracellular vesicles [157,158].

### 5.1. Neuron-Specific Enolase

Neuron-specific enolase (NSE) is a unique form of glycolytic enzyme enolase that is primarily found in neurons and neuroendocrine cells. It is considered a marker for neuronal differentiation and maturation [159,160]. In the CNS, NSE is specifically localized to neurons and is used as a marker to identify and study neuronal cells [160]. In the ENS, NSE is also expressed in neurons and neuroendocrine cells, particularly in the amine precursor uptake and decarboxylation lineage [161]. NSE has been implicated in various lung diseases and is associated with changes in cell localization and differential expression [161].

NSE levels may be altered in neurodegenerative diseases. In AD, elevated levels of NSE have been found in the CSF of patients [162,163]. Similarly, increased NSE levels in the CSF have been observed in ALS [164]. In AD, elevated serum NSE levels have been reported [165,166].

NSE has been identified in the intestinal neuronal and neuroendocrine cells of patients with CD [167]. It plays a dual role in promoting both neuroinflammation and neuroprotection in neurodegenerative events. Elevated NSE can promote extracellular matrix degradation, inflammatory glial cell proliferation, and actin remodeling, thereby affecting the migration of activated macrophages and microglia to the injury site and promoting neuronal cell death [168]. Studies have shown abnormalities in the ENS in tissue samples from patients with IBD, with an increase in neuronal cell bodies, enteroglia, and interstitial cells of Cajal in the deep muscular plexus of CD patients [169]. However, reduced NSE levels have been observed in chronic severe TBI, suggesting ongoing neurodegeneration and neuronal loss [170].

### 5.2. Neurofilament Light Chain

Neurofilament light chain (NFL) is a biomarker that can indicate neurodegeneration in various neurological disorders. NFL is a protein found in the axons of neurons, and its levels increase in the CSF and blood when there is axonal damage. There are no studies that evaluated the serum levels of NFL in IBD patients or any theory that involves its participation in the pathological pathways. Studies have shown that NFL levels are elevated in neurodegenerative diseases such as AD, PD, HD, and ALS [149,150,151].

### 5.3. S100 Proteins

S100 proteins are a family of small calcium-binding proteins that play important roles in various cellular processes such as calcium homeostasis, cell proliferation, apoptosis, differentiation, inflammation, and signal transduction. S100 proteins have been implicated in several diseases, including cancer, inflammatory disorders, neurological diseases, liver disease, and pulmonary diseases [171,172]. In the context of IBDs, S100B protein has been implicated in the onset and maintenance of inflammation in the gut [173]. It is believed to contribute to the regulation of inflammatory events in the gut and may have both trophic and toxic effects depending on its concentration [173]. In neurodegenerative diseases, S100B levels have been found to correlate with disease progression and severity [174].

## 6. Neurotrophic Factors

Neurotrophic factors are proteins that promote the survival, growth, and differentiation of neurons in the nervous system. They play a crucial role in the development, maintenance, and repair of the nervous system. Some commonly studied neurotrophic factors include nerve growth factor, brain-derived neurotrophic factor (BDNF), neurotrophin-3, glial cell-derived neurotrophic factor (GDNF), and ciliary neurotrophic factor [175,176,177].

### 6.1. Brain-Derived Neurotrophic Factor

BDNF is a protein that plays an important role in neuronal survival, growth, and neurogenesis. Changes in BDNF levels and signaling pathways have been identified in several neurodegenerative diseases, including AD, PD, and HD, and have been linked with the symptoms and course of these diseases [178]. Low BDNF levels mediate the neurodegeneration of neurons, including dopaminergic neurons in PD [179]. Moreover, BDNF improves synaptic plasticity and contributes to long-lasting memory formation [175]. Several studies have shown that a high level of BDNF is associated with a lower risk of developing a neurodegenerative disease [180]. In IBS and IBDs, BDNF plays a significant role in the modulation of abdominal pain [181,182]. BDNF expression is found to be upregulated in the colonic mucosa of IBS patients, contributing to visceral hyperalgesia [182]. IBD patients often have sleep and mood disorders, and BDNF has been shown to modulate interactions between the CNS and the gastrointestinal tract, possibly contributing to these psychological issues [183].

### 6.2. Glial Cell-Derived Neurotrophic Factor

GDNF is a protein that plays a role in promoting the survival, growth, and maintenance of various types of neurons in the CNS and PNS. GDNF is known to have neuroprotective effects on dopaminergic neurons in PD and has been studied as a potential therapeutic target for neurodegenerative diseases. It promotes the survival and maintenance of neurons, prevents apoptosis, and stimulates the growth and regeneration of nerve fibers. GDNF also modulates the phenotype and function of neurons, protects against toxic damage, and promotes the formation of new synapses [184,185,186].

Studies have shown that GDNF is upregulated in the inflamed colon of patients with IBDs and has protective effects on the intestinal epithelial barrier and enteric nervous system. GDNF has been shown to prevent the apoptosis of intestinal epithelial cells and enteric glial cells, reduce inflammation, and improve colonic transit in experimental colitis models [185,186].

## 7. Inflammatory Biomarkers

Inflammation biomarkers are measurable substances in the blood that indicate the presence and severity of systemic inflammation. They can be proteins, cytokines, chemokines, or other molecules that are released during the inflammatory response. These biomarkers can be detected in various body fluids, such as blood, CSF, or feces [187]. In IBDs, biomarkers of intestinal inflammation are important for diagnosis, disease activity monitoring, and predicting relapse. These biomarkers can help differentiate between functional and organic bowel conditions and guide treatment decisions [187]. IBDs can lead to damage to the ENS, such as plexitis or ganglionitis, which is the infiltration of immune cells into or around the ENS. Plexitis is associated with alterations in the structure and function of enteric neurons and may contribute to the progression of intestinal inflammation [188,189].

Neurodegenerative diseases are also characterized by inflammation. Neuroinflammation is a common feature of these diseases, and inflammatory biomarkers may aid in their diagnosis. Inflammatory markers in the CSF have been identified in patients with PD, LBD, and AD [50,53,54]. These biomarkers can potentially contribute to the development of diagnostic panels for these neurodegenerative diseases.

### 7.1. C-Reactive Protein

CRP is an acute-phase reactant protein produced by the liver in response to inflammation or infection. High-sensitivity CRP (hsCRP) is a more sensitive test that can detect even very low levels of inflammation and is often used to assess cardiovascular risk. Either way, measuring CRP or hsCRP provides information about the same cytokine, but hsCRP might provide more accurate information about subclinical inflammation. CRP plays one of the most important roles in the immune system by modulating the inflammatory response and promoting host defense against infections. It interacts with components of both the innate and adaptive immune systems, including complement proteins and receptors on immune cells. In the CNS, CRP is constitutively expressed by microglial cells, astrocytes, and neurons; its expression can be increased in response to inflammation and is involved in brain development, the maintenance of normal brain homeostasis, and neuroinflammation [190].

Elevated CRP levels have been observed in patients with IBD, particularly in those with CD compared to UC [143]. In a study comparing CD and UC patients, high CRP concentrations were associated with a higher odds ratio of a CD diagnosis [191]. Another study found that serum CRP levels were associated with clinical and endoscopic remission in CD patients [192]. In addition, high CRP levels were associated with poor sleep quality in patients with IBDs, independent of nocturnal symptoms [193].

There is evidence suggesting that CRP may be associated with neurodegeneration. Studies have shown that higher hsCRP levels are associated with faster declines in cognitive function, including global cognitive scores, memory scores, and executive function scores [194]. A study has found that elevated CRP levels predict poorer cognition and increased dementia risk in cognitively healthy older adults [195]. Another study showed that higher baseline CRP levels are associated with poorer memory in elderly women [196]. A population-based case-control study found that baseline hsCRP levels were significantly associated with MCI [195]. Furthermore, elevated hsCRP levels detected five years before diagnosis were associated with an increased probability of MCI [197].

### 7.2. Serum Amyloid

Serum amyloid A (SAA) is an acute-phase protein that is produced by the liver in response to inflammation or infection. SAA is involved in innate immunity by promoting the development of T helper cells and modulating various leukocyte functions. It can also bind to lipids and phospholipids, forming nanoparticles and sequestering them. In the CNS, SAA levels have been found to be increased in certain neurological disorders, such as neuromyelitis optica and atypical MS, suggesting a potential role in the pathogenesis of these conditions [198,199]. There is evidence to suggest that increased serum amyloid can transform into CNS beta plaques. Studies have shown that aggregated amyloid peptides can activate complement via the classical pathway, leading to complement activation in AD brains. Additionally, Aβ in CSF can re-enter the brain through perivascular spaces, potentially acting as a constant source of pathogenic Aβ. Furthermore, pharmacological reduction in soluble extracellular Aβ has been associated with a decrease in plaque formation and growth. These findings indicate that Aβ in the serum and CSF can contribute to the growth and deposition of beta plaques in the CNS [200,201,202].

Several studies have investigated the role of SAA as a biomarker for disease activity and mucosal inflammation in IBDs. One study found that SAA levels correlated with mucosal inflammation in patients with CD, even in those with normal CRP levels [203].

SAA has been implicated in neurodegenerative diseases. In AD, SAA has been shown to contribute to oxidative stress, mitochondrial dysfunction, impaired synaptic transmission, disruption of membrane integrity, and impaired axonal transport [204]. Additionally, SAA has been found to interact with other pathological proteins, such as beta-amyloid and alpha-synuclein, forming toxic hetero-aggregates [15,50,205].

### 7.3. IL-6

Interleukin-6 (IL-6) is a cytokine with a role in cell signaling and communication within the immune system. It is produced by various cells, including immune cells, and is involved in regulating inflammation and immune responses. IL-6 can influence the activity of different cell types, such as T cells, B cells, and macrophages. It can contribute to neurodegeneration by promoting neuroinflammatory reactions, disrupting neuron homeostasis, and inducing neuronal damage. IL-6 can also affect protein kinase pathways and tau phosphorylation patterns [206,207].

The excessive or dysregulated production of IL-6 has been implicated in the pathogenesis of several diseases, including IBDs and AD [208,209]. In a study investigating the association between IBDs and the development of AD, it was found that IBD patients had a higher risk of developing AD compared to those without IBDs. The study also found that higher levels of IL-6 were associated with an increased risk of cognitive impairment in IBD patients [70]. Additionally, another study examined the association between inflammatory markers and MCI in a population-based sample. It was found that higher levels of CRP, which is regulated by IL-6, were associated with an increased risk of MCI [210].

### 7.4. LP2-Associated Phospholipase

LP2-associated phospholipase (SPLA2) is an enzyme belonging to the phospholipase A2 family and plays a role in inflammation and host defense mechanisms against bacteria. SPLA2 also plays a role in the hydrolysis of phospholipids, specifically oxidized phospholipids found in LDL. This hydrolysis releases pro-inflammatory products such as lysophosphatidylcholine and oxidized non-esterified fatty acids, which contribute to inflammation and atherosclerosis. Elevated levels of SPLA2 have been associated with an increased risk of CVD. It is considered a potential biomarker and therapeutic target for CVD [211,212,213]. Lp-PLA2 has been found to correlate with several other blood biomarkers. In patients with coronary artery disease, Lp-PLA2 levels were found to significantly correlate with LDL cholesterol, homocysteine, and paraoxonase [214]. Another study found that SPLA2 levels were strongly correlated with total cholesterol, LDL cholesterol, and apolipoprotein B [215]. Additionally, Lp-PLA2 levels have been associated with inflammatory markers such as hsCRP [214]. These correlations suggest that Lp-PLA2 may be involved in lipid metabolism and inflammation, further supporting its role as a biomarker for CVD.

In the context of IBDs, SPLA2 has been found to have increased mass concentrations in the serum and colonic mucosa of patients with CD [216]. In patients with CD, the mass concentration of group II phospholipase A2 is increased in both serum and colonic mucosa, and it is associated with the degree of inflammatory activity in the intestinal wall [216].

SPLA2 has also been implicated in the pathogenesis of AD and MCI. For example, elevated levels of bacterial lipopolysaccharides and cyclooxygenases (COX1 and COX2) have been observed in the blood serum and CSF of patients with AD and MCI. Lipopolysaccharides positively correlate with SLPA2, β-amyloid, and tau and negatively correlate with mental state [217].

### 7.5. Prostaglandin E2

Prostaglandin E2 (PGE2) is a type of prostaglandin, which is a group of lipid compounds involved in inflammation. It is involved in regulating physiological processes, such as maintaining homeostasis, controlling vascular tone, and regulating fluid metabolism and blood pressure [218,219,220]. PGE2 also has important functions in inflammatory responses, acting as both an anti-inflammatory and pro-inflammatory mediator [219,220]. It can regulate pain perception and is implicated in the pathophysiology of vascular diseases [221,222].

In IBDs, PGE2 is involved in the control of intestinal epithelial barrier homeostasis and plays a dual role as both an anti-inflammatory and a pro-inflammatory mediator. It is produced by enteric glial cells and can help maintain gut barrier integrity [223].

PGE2 is implicated in the pathogenesis of neurodegenerative diseases. It is involved in neuroinflammation and can contribute to neuronal dysfunction induced by pro-inflammatory stimuli [224,225,226]. However, the exact role of PGE2 in neurodegenerative events is still controversial due to its anti-inflammatory actions [227].

### 7.6. IL-1β

Interleukin-1 beta (IL-1β) is a cytokine that plays a role in immune regulation and inflammation.

IL-1β has been implicated in the development and progression of IBDs. Studies have shown increased IL-1β levels in the intestines of IBD patients, and blocking IL-1β activity has been found to reduce inflammation and improve symptoms in experimental colitis models [228,229].

Elevated levels of IL-1β have been observed in the brains of AD patients and are believed to contribute to neuroinflammation and neuronal degeneration seen in the disease [230]. The relationship between IL-1β and cognitive function is complex, and conflicting results have been reported. Some studies suggest that IL-1β may impair cognitive function, while others show no effect or even a beneficial role [231,232].

### 7.7. TNF-α

Tumor necrosis factor-alpha (TNF-α) is a pro-inflammatory cytokine that activates immune responses against infection, injury, or inflammation and is involved in the regulation of various immune cells. TNF-α is also implicated in the regulation of T cell responses, including development, homeostasis, primary antigenic responses, apoptosis, effector functions, and memory cell formation. It is a key mediator of the innate immune response and has strong pro-inflammatory and immunomodulatory properties [233]. 

In patients with IBDs, increased circulating TNF-α levels have been associated with the intensity of gastrointestinal symptoms and cognitive-affective biases [234].

TNF-α can have both neuroprotective and neurodegenerative effects, depending on the context and the receptors involved. It can orchestrate immune surveillance and defense, cellular homeostasis, and protection against neurological insults under normal conditions [235]. However, in pathological conditions, such as neuroinflammation, TNF-α is released in large amounts and contributes to neurodegeneration [236,237]. TNF-α can potentiate glutamate-mediated cytotoxicity, leading to an imbalance of excitatory and inhibitory signals in the brain [237]. It can also induce oxidative stress and inflammation, which further promotes neuronal damage and neurodegeneration [236]. TNF-α can disrupt the degradation of pathological proteins, such as α-synuclein in PD and amyloid-beta in AD, leading to their accumulation [238]. Additionally, TNF-α can impair autophagic flux and compromise lysosomal function, contributing to the accumulation of toxic proteins [238].

Furthermore, TNF-α can modulate the expression of neurotrophic factors, such as nerve growth factor, which are crucial for the survival and function of neurons [238,239]. The dysregulation of TNF-α can negatively influence the synthesis of nerve growth factor and impact the development and maintenance of cholinergic neurons [238].

### 7.8. Paraoxonases

Paraoxonase refers to a family of enzymes that are involved in various biological processes, including the hydrolysis of organophosphates, aryl esters, and lactones. They encompass three isoforms, namely PON1, PON2 and PON3. PON1 is the most extensively studied member of the family and is known for its ability to hydrolyze the organophosphate pesticide paraoxon [240,241,242]. These enzymes are primarily associated with HDL-cholesterol and have been implicated in protection against oxidative stress, inflammation, and cardiovascular diseases. Paraoxonase activity has been shown to be influenced by pro-inflammatory markers such as TNF-α, leptin, IL-6, and hsCRP. Additionally, paraoxonase levels have been found to correlate with markers of oxidative stress and lipid peroxidation [243].

PON2, which is expressed in the small intestine, is involved in the antioxidative and anti-inflammatory response in intestinal epithelial cells [244]. PON1 and PON3 expression is reduced in the duodenum of celiac patients, the terminal ileum of Crohn’s patients, and the colon of UC patients compared to healthy controls [245]. PON1 and PON3 may have extracellular functions as part of the host response in IBDs and celiac disease [245]. Additionally, oxidative stress and NADPH oxidases contribute to the pathogenesis of IBDs, and PONs may be involved in regulating oxidative stress and inflammation in IBDs [246,247].

PON1 plays a role in neuroinflammation and has been associated with AD and PD [248]. PON2, on the other hand, has been shown to have a protective role against oxidative stress and neuroinflammation [249,250]. In AD, PON1 activity is reduced, and certain PON1 polymorphisms may increase the risk of the disease [251,252]. PON1 deficiency in microglia has been found to enhance microglial phagocytosis and inhibit the production of pro-inflammatory cytokines, suggesting a potential therapeutic target for AD [248]. However, the association between PON1 polymorphisms and AD development is still inconclusive [252]. In PD, PON2 has been shown to mitigate oxidative stress, enhance mitochondrial function, and exhibit anti-inflammatory properties [249].

## 8. Conclusions and Future Insights

Multiple biomarkers were reviewed, but only some of them proved to be useful clinical tools. Some confounding factors might require attention when testing the hypothesis of whether neurodegeneration is a consequence of IBDs, such as therapy effect, different comorbidities, malnutrition, and the effect of genetic and environmental factors (Figure 2).

Vitamin D3 plays an important role in the pathophysiology of both IBDs and neurodegenerative diseases. Studies indicate that vitamin D3 deficiency is common in patients with IBDs and may worsen disease activity. Supplementation has shown promise in improving outcomes, including reducing the risk of relapse and surgeries. Correcting vitamin D3 levels could potentially mitigate inflammation and neurodegeneration in IBD patients. An objective should be maintaining a healthy level of vitamin D3. Studies that evaluate the exact mechanisms in the CNS could help us obtain more information about the direct effects of vitamin D3 on CNS structures. In addition, some prospective studies should investigate whether and how the low levels of vitamin D3 correlate with the development of neurodegenerative diseases. Studies that evaluate the benefits of vitamin D3 supplementation in neurodegenerative disease prevention could also be carried out.

Vitamin B12 might play a role in the pathophysiology of IBDs and neurodegenerative diseases. The deficiency of vitamin B12 is prevalent in patients with IBDs, particularly in those with CD. Its deficiency is associated with cognitive impairment, and supplementation has shown benefits in cognition for those with pre-existing deficiencies. Further studies are needed to determine the efficacy of supplementation in preventing neurodegenerative diseases in IBD patients. Healthy levels of vitamin B12 should be maintained before the development of a neurodegenerative disease as it may be helpful.

A smaller role was found for vitamin B9 in the pathophysiology of IBDs and neurodegenerative diseases. However, the serum levels of vitamin B9 must be tested in patients who have elevated levels of homocysteine [253]. While folate plays important roles in various biological processes, including neurodegeneration, its exact role in immunity, inflammation, and IBDs is still not fully understood. Further research is needed to elucidate the mechanisms and potential therapeutic implications.

HHCY is associated with the development of neurodegenerative diseases through various mechanisms. Testing the serum levels of HCY should be performed in patients diagnosed with IBDs because it is an important cardiovascular risk factor as well as a predictive factor for neurodegenerative disease development. Elevated homocysteine levels are linked to cognitive decline and an increased risk of neurodegenerative diseases. Supplementation with folic acid and/or vitamin B12 has shown potential benefits in improving cognitive performance, suggesting a potential avenue for intervention in IBD patients to mitigate neurodegeneration.

Thiamine and pyridoxine play a role in the CNS, including energy production, neurotransmitter synthesis, and immune function. In the context of IBDs, vitamin B1 and B6 deficiencies may occur, and supplementation may be necessary. In neurodegenerative diseases, vitamin B deficiency has been associated with cognitive impairment, but the long-term effects of supplementation are still uncertain. Clinical trials investigating supplementation, particularly in CD patients with malabsorption, could shed light on their potential neuroprotective effects.

Neuron-specific enolase could be modified in IBD patients in either way. Studies should be carried out to see how these diseases affect its serum level.

NFL might be used as a diagnostic, prognostic, and monitoring biomarker for neurodegenerative diseases. While NFL can indicate the presence of neurodegeneration, it does not provide information on the underlying cause, necessitating additional diagnostic tests [148].

In a similar manner to NFL, S100 proteins can indicate the presence of neurodegeneration, but it does not provide information on the underlying cause. High levels of S100B are suggestive of pathogenic processes in these diseases and may be involved in the activation of inflammatory pathways [254].

While neuronal damage biomarkers show promise in the context of neurodegenerative diseases, their utility in assessing neuronal damage specifically related to IBDs requires further research. Investigating these biomarkers in the context of IBDs may provide valuable insights into the interplay between gut inflammation and neuronal dysfunction, potentially leading to improved diagnostic and therapeutic strategies for IBD patients.

Currently, neuronal growth factors, as biomarkers, do not seem useful due to their lack of specificity.

BDNF has been linked to neuronal survival and neurogenesis, and its dysregulation has been observed in various neurological disorders. While its role in IBD-related neuronal damage is not well understood, its involvement in modulating abdominal pain and psychological symptoms in IBD patients warrants further investigation.

GDNF and BDNF could be useful for prospective studies involving the development of neurodegenerative diseases in IBD patients. Their clinical use is now limited because they lack specificity, but future research might change this affirmation.

Monitoring inflammatory biomarkers in the context of IBDs is essential for diagnosis, disease activity monitoring, and treatment decisions. Biomarkers like CRP, IL-6, TNF-α, and others help in assessing the severity of inflammation and guiding therapeutic interventions. However, in the case of neurodegenerative diseases and MCI, the utility of monitoring inflammatory biomarkers is less clear. While neuroinflammation is a common feature of neurodegenerative diseases, the relationship between systemic inflammatory biomarkers and the development or progression of these conditions is complex and not well-established. While some studies have shown associations between inflammatory biomarkers like CRP, IL-6, and TNF-α with cognitive decline and neurodegenerative diseases, their utility as predictive factors remains uncertain.

A causal relationship between CRP levels and neurodegenerative diseases was not proven by any of the studies. There was no study that evaluated the serum levels of CRP and the development of MCI or any other neurodegenerative disease in the IBD population. Associated with other biomarkers, CRP might be useful for monitoring the development of neurodegenerative diseases.

SAA might play a role in the pathogenesis of neurodegenerative diseases. A proposed mechanism, which involves elevated serum levels of SAA that transform into beta plaques in the CNS, could be investigated in IBD animal models. More research is needed to fully understand the mechanisms and potential therapeutic implications of SAA and to test this hypothesis.

IL-6 has been studied as a molecule linking IBDs to the development of neurodegenerative diseases. There are multiple theories on how it influences cerebral function, but none of them is conclusive. Serum levels of IL-6 should be monitored to see whether there is a correlation or a causative role between the increase in IL-6 levels and the development of neurodegenerative diseases.

SPLA2 and lipopolysaccharides might play a role in the pathogenesis of neurodegeneration due to increased CVD risk, which increases the risk of vascular dementia and increased neuroinflammation. SPLA2 is correlated with CRP levels and serum lipid levels; so, monitoring them might be useful in the long term to avoid the additional costs of dosing SPLA2, which is an uncommon blood analysis. Paraoxonase is also correlated with inflammation biomarkers and lipid levels; therefore, their clinical use is costly.

The specific relationship between PGE2 and neurodegeneration is not well-established; it is possible that PGE2 might play a role in neuroinflammation and neuronal dysfunction associated with cognitive impairment. On the other side, PGE2 also has anti-inflammatory effects, which is contrary to the theory that it is a molecule implicated in the development of neurodegenerative diseases. IL-1β also plays a dual role in IBDs and neurodegenerative diseases, and no conclusion has been drawn. Neither PGE2 nor IL-1β should be tested as they lack clear mechanistic evidence now.

TNF-α also plays a dual role by promoting and protecting against neurodegenerative diseases. Anti-TNF-α treatment is quite commonly used in IBD patients. Prospective studies that clinically evaluate the serum levels of TNF-α correlated with cognitive scores should provide insight into whether it is beneficial or not to diminish the circulating levels of this molecule.

IL-1β is important in inflammatory responses and has been implicated in both IBDs and cognitive impairment. Conflicting results were reported. The clinical relevance is low, and this molecule is not worth testing.

Paraoxonase serves a dual role, similar to most inflammation biomarkers. Their correlations with other inflammatory biomarkers and serum lipid levels should be monitored in the evolution of IBDs in patients.

Therefore, while monitoring inflammatory biomarkers may be valuable for assessing disease activity and guiding treatment in IBDs, their utility in predicting or monitoring neurodegenerative diseases like AD or PD is currently limited. Future research may provide more insights into the relationship between systemic inflammation and neurodegeneration and help identify more specific biomarkers or therapeutic targets for these conditions.

Neuronal damage biomarkers hold promise for various neurological disorders, including neurodegenerative diseases. However, their utility in the context of IBDs is less established.

NSE is a marker for neuronal damage, but its elevation can also be influenced by factors like intestinal inflammation. Therefore, its specificity for detecting neuronal damage in IBDs may be limited. Further research is needed to understand its role in the context of IBDs.

NFL is a promising biomarker for neurodegeneration, but its application in IBDs has not been explored. Given its potential to indicate axonal damage, it may be worth investigating its utility in monitoring neuronal integrity in IBD patients.

S100 proteins have been implicated in both IBDs and neurodegenerative diseases, suggesting a potential link between gut inflammation and neurodegeneration. However, their specific role in IBD-related neuronal damage requires further investigation.

To explore biological links between the gut and the brain regarding the development of neurodegeneration as a complication of IBDs, several pathways of research should be followed:Exploring the common pathways: (1) changes in the gut microbiome and metabolites that are secreted by the gut microbiome, which bypass the intestinal barrier, reach the bloodstream, and finally cross over the BBB and (2) systemic inflammation and how it affects the cerebral circulation. The important aspect of this is to identify whether the structural changes in the brain are the effects of IBDs or the cumulative effects of inflammation and comorbidities.Inflammatory pathway: research regarding correlations between the inflammatory biomarker levels and the development of neurodegenerative diseases.Therapeutic interventions regarding the effects of probiotics and whether they could prevent the development of neurodegenerative diseases. Interventional studies focusing on dietary modifications or supplementation to increase vitamin levels could provide valuable insights. However, patients with malabsorption issues may require parenteral supplementation. Additionally, strategies such as probiotics to enhance vitamin production in the gut microbiome could be explored.Neuroimaging: the serum levels of biomarkers should be correlated to the degree of cerebral atrophy or metabolism, but this requires prospective studies. This hypothesis, which is very expensive to explore, should be tested soon.Lifestyle factors associated with IBDs: different diet regimens could potentially influence the gut microbiome, thus influencing the development of neurodegenerative diseases. Different diets might also change the serum levels of vitamins, which have been linked to neurodegeneration.Genetic risk: patients diagnosed with IBDs should be questioned for the family history of neurodegenerative diseases to be included in prospective observational studies to test the hypothesis of whether there are common genetic factors between IBDs and neurodegenerative diseases.Medication effects: observational prospective studies must be conducted regarding the development of MCI and neurodegenerative diseases in patients with IBDs using different disease-modifying therapies.

## Figures and Tables

**Figure 1 ijms-25-05676-f001:**
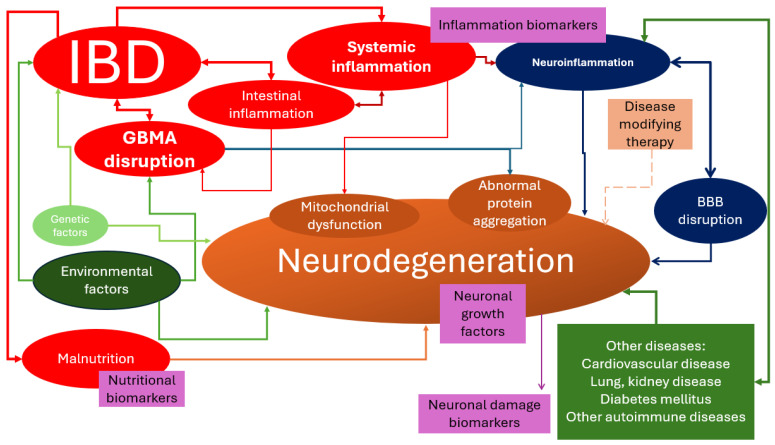
Presumed mechanisms of neurodegeneration and the place of serum biomarkers in patients diagnosed with IBDs (IBDs encompass GBMA disruption and intestinal and systemic inflammation, which develop a pathogenic cascade, leading towards neurodegeneration).

**Figure 2 ijms-25-05676-f002:**
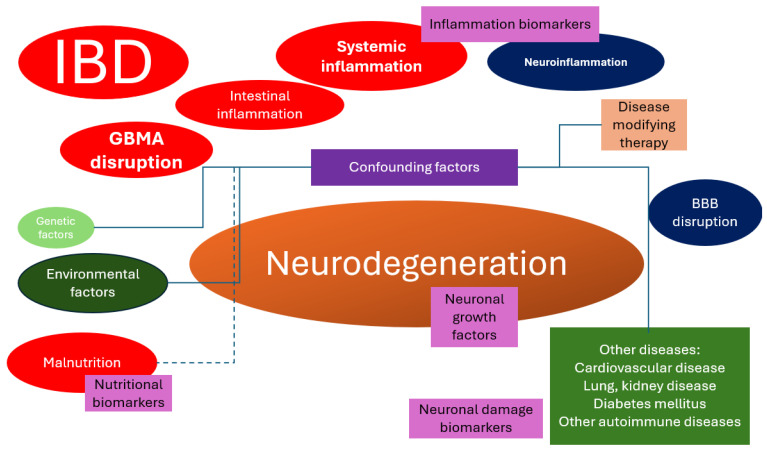
Presumed biomarkers of neurodegeneration in IBD patients (confounding factors that could contribute to false results are genetic factors, environmental factors, possibly malnutrition, disease-modifying therapy, and other diseases). Continous lines should involve a stronger correlation while dashed lines involve a lesser correlation.

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
