# Peer review of "From Gut to Brain: Uncovering Potential Serum Biomarkers Connecting Inflammatory Bowel Diseases to Neurodegenerative Diseases"

_ijms, 2024, doi:10.3390/ijms25115676_

Round 1

Reviewer 1 Report

Comments and Suggestions for Authors

General

This review addresses the topic of possible links between various types of cognitive impairments and a number of immune, microbial, biochemical, and metabolic factors in patients with IBD as reported in the literature. The topic is not new, but the manuscript is written and referenced rather exhaustively. The structure of the review is less than ideal as factors are not mentioned in a logical or organized fashion, and the quality of each section is uneven. At the end of the manuscript there is no clear final message so that the reader can take home the notion of whether the multiple reports and correlations listed in the review are valid or specific for IBD.

Major comments

Page 1, Title.

The title is incorrect and misleading. Only “mild cognitive impairment” is mentioned when in reality the postulated “molecular links” discussed throughout the manuscript are associated with several other forms of central nervous system ailments.

Page 1, Abstract

Some of the words like “leading to” and “explore the links” suggest a direct cause-and-effect relationship between any specific factor and reported cognitive impairments, which is not correct. This must be changed to more correctly suggest an association or correlation, and not an established cause-and-effect. In fact, all the postulated relationships between the factors listed throughout the manuscript are nothing more than associations without a demonstrated molecular explanation, and this should be made evident to the readers.

Page 2, Aim

The various forms of the cited cognitive impairments, i.e., MCI, NDDs, AD, and PD are not defined, and they all may be mild, moderate or severe.

Page 3

In the second paragraph of the third section starting with “Some studies have shown…” should be in the introduction and should point out that not only IBD but many other immune-mediated inflammatory conditions (IMIDs) are also associated with cognitive impairment.

Page 4

In the first paragraph of the second section starting with “MCI is a condition…” it is not clear whether the “individuals” are IBD patients. 

In the same paragraph “a pathogen” is mentioned: what pathogen?

Page 4

“Serum biomarkers can serve as predictors for MCI”. This is not true. They have been studied and some associations found, but an actual proven and reliable predictive value has never been established.

Page 5-9

When a specific factor (CRP, vit. B12, homocysteine, amyloid, enolase, neurofilament, S100, neurotrophic factor, IL-6, LP-2, PFG2, IL-1β, and TNF-𝝰) is mentioned and introduced at the beginning of each section a key reference supporting the definition and known function must be introduced.

The fact that there are no studies of most of the specific factors associated with IBD should be made clear in an initial separate paragraph rather than at the end of each paragraph. It should also be made clear that associations between each factor and some form of central nervous system impairment have been reported, but actual cause-and-effect-links are yet to be proven and data are inconclusive.

Page9

There is no mention to IBD when PGE2 is discussed.

Page 10, Conclusions

The conclusion paragraph is too succinct, poorly written and uninformative. What is the final message of the review? What perspective or insights does the review offer? What are the recommendations for the future? What needs to be done to establish relevant and credible connections between each factor and cognitive impairment instead of simple associations? It should also be made clear that no actual “molecular links” of any type have been properly investigated and subsequently established, and that the literature only reports simple associations and not mechanistic correlations.

Additional comments

Page 1, Abstract

All abbreviations mentioned in the abstract are not explained, which is confusing for the reader. They should all be spelled out.

The abbreviation list should be at the beginning of the manuscript.

The sentence “current results are lacking clarity” is inappropriate. Please replace with “they are inconclusive”.

Page 10, Conclusions

“studies” should be “studied”.

Comments on the Quality of English Language

Some improvements can be done

Author Response

Dear Mrs/Ms/Mr,

We hope this message finds you well and we thank you for the comments made to our review.

We agree that the structure was not done ideally and that conclusions were rather lacking in the manuscript.

In the updated form, these issues were solved, hopefully.

Regarding the comment list, here are our point answers:

  1. Title was changed as it was suggested also by the other reviewer.
  2. Abstract was corrected and it’s more accurate now.
  3. The aim was changes and we defined what was needed for a better unstanding.
  4. We agree to those wrong causal relantionships we have created and we have solved.
  5. We created less sections in order to reduce the confusion of the reader.
  6. Conclusions are now more robust and fulfull the aim and methodology.
  7. Abreviation list was placed at the end as we didnt find it fitting at the start. But if required, we could change it.

Some of the changes had to be in agreement with the other’s reviwer’s opinion.

We await your next review,

Best regards,

Oliviu Sarb and the collective of authors.

Reviewer 2 Report

Comments and Suggestions for Authors

This article is a review focusing on the complex interplay between inflammatory bowel disease —including Crohn's Disease and Ulcerative Colitis —and its extraintestinal neurobiological manifestations, notably mild cognitive impairment and neurodegenerative diseases. In general strengths of the manuscript include an interdisciplinary approach. The manuscript bridges gastroenterology and neurology, supplying a multidimensional overview of how gastrointestinal health affects neurological conditions. Additionally, it comprises a wide range of biomarkers, suggesting a thorough investigation of the existing literature. The main weakness is ambiguity in biomarker usefulness. Despite discussing various biomarkers, the manuscript notes the current results lack clarity, which might leave readers without definitive guidance on the reliability of these markers.

A few suggestions to improve the manuscript are made here:

1. The manuscript lacks a real mechanistic approach. It is written very briefly and does not delve into the mechanisms underlying the diseases discussed (IBD and cognitive impairment). For the article to effectively explain the molecular links, it is crucial to provide a detailed explanation of the pathophysiology at the organism, cellular, and molecular levels. This is a significant omission that needs to be addressed to enhance the depth and understanding of the review.

2. The structure of the article further demonstrates its lack of depth in the mechanistic approach. It is divided into many short subsections, resulting in repetitive information. I suggest consolidating the descriptions of related biomarkers, such as vitamins and pro-inflammatory cytokines, to enhance clarity and coherence. To enhance the manuscript's clarity and coherence, it would be beneficial to consolidate the information about related biomarkers. This could involve grouping biomarkers that share similar functions or effects. A systematic organization could include sections that detail each group's implications for IBD and its extraintestinal manifestations, particularly cognitive functions. This approach would provide a more structured and comprehensible overview, allowing readers to better understand the connections and distinctions among the various biomarkers discussed.

3. The title of the manuscript does not accurately reflect its actual content. I suggest a title that indicates the authors' focus on describing potential biomarkers linking IBD and cognitive impairment. I propose something like: "Identifying Biomarkers: Exploring the Link Between Inflammatory Bowel Disease and Cognitive Impairment" or "From Gut to Brain: Uncovering Biomarkers that Connect Inflammatory Bowel Disease to Cognitive Dysfunction".

4. Abbreviations should be explained and introduced before they are used. The abstract contains numerous abbreviations that are not defined, which can be confusing for the reader.

5. Epidemiological data on both IBD and cognitive impairment should be presented to provide a comprehensive background for the study.

6. The description of the impact of IBD on the brain and behavior should be expanded upon by including a detailed discussion of the mechanisms involved.

7. The Aim and Methodology sections could be combined to streamline the presentation and improve the flow of information in the manuscript. Additionally, I do not see the necessity for Diagram A. It presents a methodology of research that is quite simple and could be adequately described in the text itself.

8. The paragraphs discussing neurodegeneration as an extraintestinal manifestation of IBD and Mild Cognitive Impairment—including definitions, risk factors, and diagnosis—lack depth. Please expand these sections to include a more detailed elucidation of the pathological changes that occur, with a focus on the cellular and molecular levels. This will help to provide a clearer understanding of the link between IBD and cognitive changes.

9. Some definitions, such as the GBA axis, are repeated multiple times throughout the manuscript, which is unnecessary. Additionally, in the paragraph discussing the presumed link between IBD and MCI, it would be beneficial if the authors could specify which brain structures are particularly affected.

10. The authors describe the results of other studies quite briefly, often without providing broader context or essential details. For example, on page 4, in the paragraph titled "Gut Microbiome Alteration and MCI," when citing reference 30, the authors fail to specify the pathogen on which the tests were performed. Additionally, the descriptions of various markers frequently lack basic information, such as the specific tissues where the concentrations of the tested substances were found to be increased or decreased. This information is crucial for understanding the relevance and implications of these findings.

11. The mechanism of vitamin D action on IBD symptoms and GBA should be provided in the manuscript. This would help clarify how vitamin D influences these conditions at a moleluclar level.

12. The description of biomarkers in the manuscript is chaotic. The narrative often shifts abruptly, starting with effects on the CNS, moving to CD, then back to the CNS, and finally to IBD more broadly, as seen with biomarkers like serum amyloid and neuron-specific enolase and many others. This disjointed approach detracts from the readability of the manuscript. It would be beneficial to organize these descriptions in a more systematic and cohesive manner, perhaps by grouping biomarkers by their primary effects or areas of impact.

13. Including a schematic iconography that presents the pathogenesis of IBD and its implications on cognitive functions would be beneficial for the review. This should also depict where proposed biomarkers are involved in the process, providing a clear visual representation that enhances understanding of the complex interactions.

14. In the conclusion, the authors should address the practical application of the proposed biomarkers. Specifically, they should discuss which biomarkers can be easily measured in serum or plasma and thus could be introduced into clinical practice. This would enable monitoring and potentially reduce the risk of MCI development in patients with IBD. It would be helpful for the authors to highlight the feasibility and benefits of using these markers in routine clinical settings to improve patient outcomes.

15. The authors also mentioned that "The objective is to identify potential correlations and propose future research directions to understand the impact of altered microbiomes and intestinal barrier functions on NDDs." However, the manuscript lacks specific details about these future research perspectives. It would be beneficial for the authors to outline potential studies or methodologies that could further explore these correlations, including experimental designs, potential cohorts, and biomarkers that could be investigated. This addition would not only fulfil the stated objective but also provide a clear path forward for the field.

Author Response

Dear Mrs/Ms/Mr,

We hope this message finds you well and we thank you for the comments made to our review.

Regarding your suggestions, updates were done, point by point:

  1. Mechanisms and molecular links were added in order to be accurate to the title.
  2. The strucuture was simplified and biomarkers were grouped by similarity.
  3. Title was changed to reflect better the content.
  4. Abstract was changed in order to remove the confusing abreviations. All abreviations are enlisted and explained now in the text.
  5. We agree that some epidemiological data should have been added so it was done accordingly.
  6. The impact of IBD on brain and behaviour was added.
  7. Aim and methodology were combined and text was simplified and its now more explicit were it lacked.
  8. We have deleted the paragraphs about MCI because they were not really the object, rather the explications in the introduction are enough. Mechanisms were added.
  9. The repeated definitions were removed if multiple. Affected brain structures are now mentioned.
  10. Yes, we now mentioned were the biomarkers are mentioned where the info was missing.
  11. The mechanism of vitamin D3 on IBD simptoms and GBA was added.
  12. The description is now clear.
  13. Schematics for pathogenesis were added (Fig 1 and 2)
  14. Practical solutions were expanded now in the conclusion.
  15. Future research directions were added.

Best regards

Oliviu Sarb and the collective of authors.

Round 2

Reviewer 1 Report

Comments and Suggestions for Authors

General

The revised version of the manuscript is substantially different from the original one and significantly improved in regard to structure, presentation, flow, message and conclusions. However, it can be further improved by addressing the relatively minor issues listed below.

Page 1, title

The title is better now, but it is still not fully reflective of the manuscript message, and the following title is suggested: “From gut to brain: uncovering potential links connecting inflammatory bowel disease to cognitive disorders”.

Page 1, abstract

Delete “clinical response to probiotics”.

Page 2, second section

Explain “protein aggregation”.

Page 3, top paragraph

Replace “deceiving” with “disappointing, discouraging, inadequate or unsatisfactory”

Page 3, third paragraph

Replace “side” with “hand”.

Page 3, third paragraph

Replace “cause” with “consequence”.

Page 5, second paragraph

Explain “default mode network”.

Page 9, fifth paragraph

Break the sentence starting with “These biomarkers..., ways”. Place a comma after “ways” and star a new sentence: “For example, they can…”.

Page 9, fifth paragraph

Delete “non-invasive”. Serum biomarkers may not be invasive for the brain, but they still require an invasive procedure, i.e., blood sampling.

Page 9, sixth paragraph

What is the connection of ENS with cognitive impairment? The ENS is relevant to IBD, but it is not clear in regard to the CNS. Please clarify and change.

Page 10, first paragraph

Change to ENS or CNS”.

Page 10, third paragraph

Replace “It” with NSE.

Page 10, neurofilament light chain

No mention to IBD. If this is so, state it.

Page 10, S100 proteins

“Same as NFL, the potential use resides from the same reasoning”. What does this mean? Please rephrase.

Page 10-11, C reactive protein

CRP and hsCRP are the same thing, they both measure CRP. The way that this section is written gives the impression that they are two different things, which is incorrect. Just mention CRP, the way it is measured does not matter.

Page 14, point 3.

Replace “installation” with “development” or “induction”.

Comments on the Quality of English Language

Acceptable with a few exceptions mentioned in the review

Author Response

Dear Mrs/Ms/Mr,

We hope this message finds you well and we thank you for the comments made to our review.

In the updated form, these issues were solved, hopefully.

The review was drastically changed, so point answers are hard to be done.

Some of the changes had to be in agreement with the other’s reviwer’s opinion.

We await your next review,

Best regards,

Oliviu Sarb and the collective of authors.

Reviewer 2 Report

Comments and Suggestions for Authors

The article has been revised, but I am dissatisfied with the outcome as it appears my comments were not adequately addressed. The authors' responses, such as 'Mechanisms and molecular links were added' or 'The impact of IBD on brain and behavior was added,' lack specific details on the changes made. This lack of specificity suggests that there was insufficient effort to fully incorporate the feedback provided.

The authors continue to describe the proposed mechanisms quite briefly (lines 137-156), with some repetition noted (e.g., items 2 and 4).

In the newly created sections, the authors frequently cite a substantial amount of literature collectively over just a few sentences—for example, in lines 211-216, nine references are cited across three sentences with citations placed at the end of the paragraph, similar to lines 194-200. This approach makes it unclear which specific pieces of information are derived from each source.

The authors have indicated a shift in focus from MCI to dementia, specifically Alzheimer's Disease (AD) and Parkinson's Disease (PD), and removed content related to MCI. However, I observe no substantial changes reflecting this shift. Additionally, I recommend that the pathophysiology of changes in AD and PD be described in more detail to adequately support the new focus of the article.

The article lacks a detailed mechanistic approach and suffers from structural incoherence. For instance, the authors mention nutritional deficiencies as a significant issue for IBD patients and then proceed to describe individual vitamins as markers without a connecting narrative. Notably, there is no explanation or transition that justifies why these vitamins, such as Vitamin D, can be considered as biomarkers, creating a gap in the logical flow of the text.

Most of my comments from the previous review, specifically comments 1, 5, 6, 8, 10, and 12, still apply to the current version of the manuscript submitted for review.

Comments on the Quality of English Language

The newly added sections of the manuscript exhibit issues with linguistic accuracy, making them more challenging to read.

Author Response

Dear Mrs/Ms/Mr,

We hope this message finds you well and we thank you for the comments made to our review.

Here are some point answers to your comments regarding our review.

  1. A big rewriting was implemented so the structure was changed, focusing now only on mechanisms in the first part, then undertaking each biomarker with some specific mechanisms at each one.
  2. References were cleared and placed correctly and they have been divised for each sentence (where it was needed).
  3. The shift was made towards neurodegeneration, as we felt we covered very much of it even in the first review.
  4. We added pathophisiology so the readers understand this review much better.

We hope that this new version is fit for the journal, with language checks.  

Best regards

Oliviu Sarb and the collective of authors.

Round 3

Reviewer 2 Report

Comments and Suggestions for Authors

Due to the significant improvement of the manuscript, I recommend its publication.